# 0D van der Waals interfacial ferroelectricity

Yue Niu[1,2,5], Lei Li[1,2,5], Zhiying Qi[1,2], Hein Htet Aung[1,2], Xinyi Han[1,2], Reshef Tenne [3], Yugui Yao [1,2], Alla Zak [4] & Yao Guo[1,2] ✉

The dimensional limit of ferroelectricity has been long explored. The critical contravention is that the downscaling of ferroelectricity leads to a loss of polarization. This work demonstrates a zero-dimensional ferroelectricity by the atomic sliding at the restrained van der Waals interface of crossed tungsten disufilde nanotubes. The developed zero-dimensional ferroelectric diode in this work presents not only non-volatile resistive memory, but also the programmable photovoltaic effect at the visible band. Benefiting from the intrinsic dimensional limitation, the zero-dimensional ferroelectric diode allows electrical operation at an ultra-low current. By breaking through the critical size of depolarization, this work demonstrates the ultimately downscaled interfacial ferroelectricity of zero-dimensional, and contributes to a branch of devices that integrates zero-dimensional ferroelectric memory, nano electro-mechanical system, and programmable photovoltaics in one.

The scaling-down of ferroelectricity has been long pursued in the past decades[1–5], which is the fundamental of miniaturization and integration of ferroelectric devices. The central challenge, in the presently accepted paradigm, is a size effect: down-scaled ferroelectricity diminishes due to the arising depolarization field. Advances in low-dimensional material preparation, characterization, and fabrication have promoted the vigorous development of nanoscale ferroelectrics[6,7]. For example, the recent research on two-dimensional (2D) ferroelectricity has brought the thickness down to the atomic limit, including the ultrathin doped hafnium/zirconium oxide[8–10], the monolayer ferroelectric materials[11–15], and the van der Waals (vdW) stacked assembly[16–30]. Despite the success of 2D ferroelectricity, trials to further scale down the ferroelectricity of the solid state, for example, to the ultimate zero-dimensional (0D) ferroelectricity, have been very limited[31]. Recently, the vdW interfacial ferroelectricity, or sliding ferroelectricity of stacked hexagonal boron nitride, transition metal dichalcogenides (TMDCs), and sandwiched graphene enabled the design of ferroelectric systems out of non-ferroelectric parent compounds[16–30], expanding the scope of ferroelectric materials from the wide-band insulators to semiconductors and metals. Given the demonstration by the 2D stacked vdW assemblies, it is imperative to investigate if such a unique form of ferroelectricity could break the limitation of scaling down and build the ultimate 0D ferroelectrics in the solid state.

Ferroelectrics are important candidates for nonvolatile memory as part of the quest for denser storage, lower power consumption, and neuromorphic computing[32–36]. Among the ferroelectrics, the ferroelectric diode is a resistance-switching device[37,38] that allows operation at a current several orders of magnitude smaller than nano electro-mechanical system (NEMS) memory (>$10^{-2}$ A)[39], phase change random access memory (PcRAM, >$10^{-5}$ A)[40], resistive random access memory (RRAM, $10^{-4}$ A)[41], magnetoresistive random access memory (MRAM, $10^{-3}$ A)[42] without extra restrictor. While the ferroelectric diode also generates a switchable photovoltaic effect, the sensitive wavelength range, however, is limited to the high photon energy due to the wide band gap of the traditional ferroelectric materials[43]. Nevertheless, the integration of ferroelectric diode memory and switchable photovoltaics based on the vdW interfacial ferroelectricity has not yet been demonstrated. One preventive reason could be that the vdW interfacial ferroelectricity in 2D contains multi-domains that could hinder the overall polarization, urging efforts in downscaling the vdW interfacial ferroelectricity within one single domain.

This work realizes the scaling of the vdW interfacial ferroelectricity down to 0D (<10 nm × 10 nm × 2 nm), breaking the dimensional

[1]Centre for Quantum Physics, Key Laboratory of Advanced Optoelectronic Quantum Architecture and Measurement (MOE), School of Physics, Beijing Institute of Technology, 100081 Beijing, China. [2]Beijing Key Lab of Nanophotonics & Ultrafine Optoelectronic Systems, School of Physics, Beijing Institute of Technology, 100081 Beijing, China. [3]Department of Molecular Chemistry and Materials Science, Weizmann Institute of Science, 7610001 Rehovot, Israel. [4]Faculty of Sciences, Holon Institute of Technology, 52 Golomb Street, 5810201 Holon, Israel. [5]These authors contributed equally: Yue Niu, Lei Li. ✉e-mail: yaoguo@bit.edu.cn

limitation of depolarization. By stacking crossed one-dimensional tungsten disulfide ($WS_2$) nanotubes, the created 0D interface results in a spontaneous electric polarization switch via vdW sliding. The constructed four-probe device exhibits not only bipolar nonvolatile resistive switching at low operation current, but also the programmable photovoltaic effect in the visible band covering the spectral range from red to blue. Here we demonstrate the vdW interfacial ferroelectricity downscaled to the ultimate 0D, and contributes a branch of interfacial ferroelectric diode device that combines the concepts of 0D ferroelectricity, semiconductor ferroelectrics, and NEMS memory for the high-density low-power memory and photovoltaic switch.

## Results and discussion
### Building the 0D vdW interface
The first step towards the 0D interfacial ferroelectricity is to construct the vdW interface within a small area. As shown in Supplementary Fig. 1, the feasible strategy to construct a 0D vdW interface includes contacting the 0D vdW component to another, or assembling the 1D vdW components in a cross[44,45]. The latter, with four extended arms, is obviously more suitable for further device fabrication, characterization and further integration. In this work, we assemble the crossbar of $WS_2$ nanotubes (see charaterizations of $WS_2$ nanotubes in Supplementary Figs. 2 and 3) using the clean dry transfer technology[46], and fabricate the devices with the four terminals contact electrodes, as shown schematically in Fig. 1a and Supplementary Fig. 4. Scanning electron microscopy (SEM) and atomic force microscopy (AFM) images of a fabricated device are shown in Fig. 1b, c. To evaluate the contacting area of the interface, we feed the heights (diameters) of the nanotubes ($h_1$, $h_2$) and the height of cross junction ($h_3$) to the finite element model to simulate the deformation, the pressure, and the scale of the vdW interface. As shown in Fig. 1d and Supplementary Fig. 5, the deformation indicates a nummular interface of about

10.0 nm by diameter, or 78.8 $nm^2$ by area ($h_1 + h_2-h_3 \sim 1.0$ nm). This is 178 times smaller than the size of a single domain of 2D vdW interfacial ferroelectricity, as illustrated in Supplementary Fig. 6. The results also indicate a maximum compacting pressure intensity of up to 2.24 GPa, or $2.21 \times 10^4$ times of atmospheric pressure, due to the limited contacting area. Such a large pressure intensity enhances the vdW coupling at the interface and contributes to a higher current density and coercive field, as will be discussed with the results below. With such 1D−1D $WS_2$ nanotube crossbar configuration, we obtain the 0D vdW interface in the middle of the semiconductor components.

We used the four-terminal measurement to characterize the electrical performance of the 0D interface, as shown in the insert of Fig. 1e and Supplementary Fig. 7. Unlike 2D vdW assembly that can expose the electric potential to the scanning probe microscope or the channeling contrast electron microscope[19,21,22,25−27], the 0D interface is secluded and its properties have to be presented through electrical characterization. The four-terminal measurement ensures that the $I$-$V$ curve in Fig. 1e consists of the current and voltage drop just across the 0D interface. The $I$-$V$ curve show nonlinearity and the current decreases upon decreasing temperature, indicating a potential barrier for the carriers at the 0D junction. An equivalent barrier height of the 0D junction as a function of voltage was extracted from the Arrhenius plot of the current-temperature relationship, as shown in Fig. 1f and Supplementary Fig. 8[47,48]. The equivalent barrier should be from the coupling between the vdW interface plus the slight difference between the band structure of the two nanotubes of a different diameter[49]. The equivalent barrier height is above one hundred meV and decreases with the voltage drop across the 0D vdW interface. Among all the fabricated devices, we note that although the junctions are with the same $WS_2$ nanotube components besides the 0D interface, the $I$-$V$ curves could be asymmetric and shows different levels of rectification, indicating an interfacial polarization across 0D homojunction.

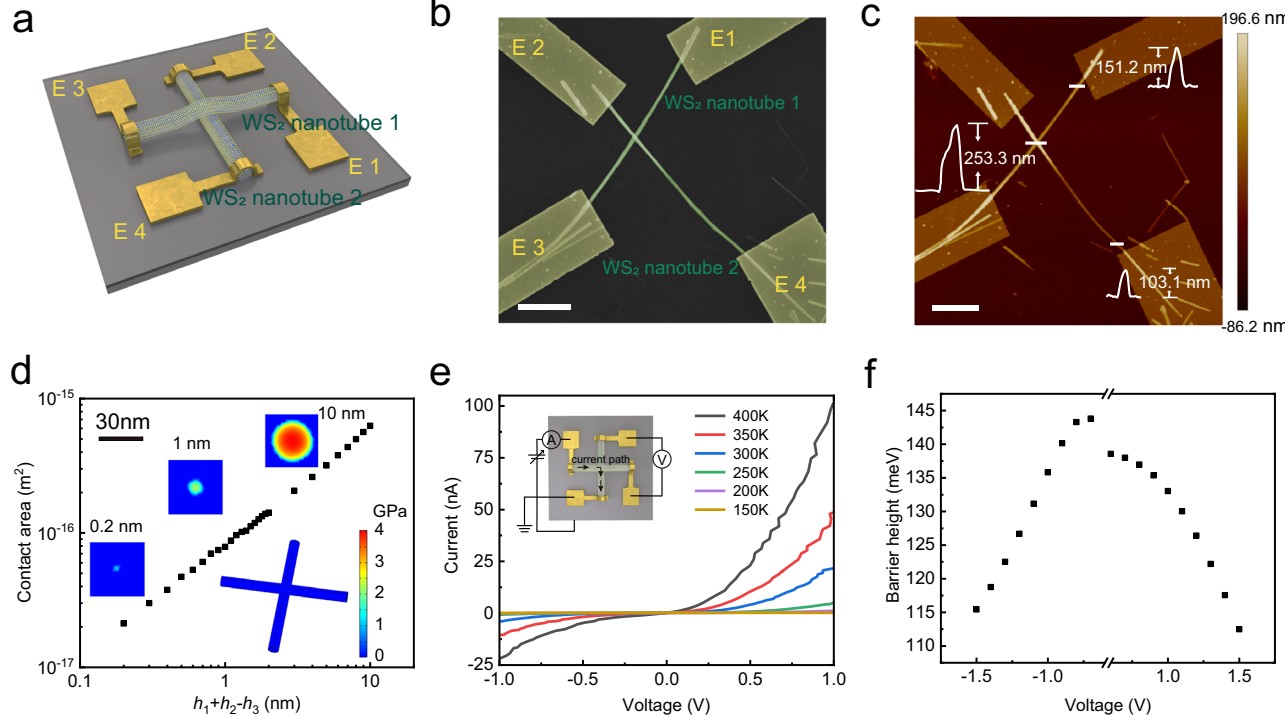

**Fig. 1 | The structure and electronic property of $WS_2$ nanotubes crossbar.** **a** Schematic diagram of the device structure. **b** SEM image of the $WS_2$ nanotubes crossbar. The scale bar is 3 µm. **c** AFM image of the $WS_2$ nanotubes crossbar with the heights $h_1$, $h_2$, and $h_3$ labeled. The scale bar is 3 µm. **d** The relationship between the interface area and the height difference $h_1 + h_2 − h_3$. Insert: Simulated $WS_2$ nanotubes crossbar structure and the pressure distribution. The Young's modulus is extracted from ref. 81,82. **e** $I$−$V$ characteristics of $WS_2$ nanotubes crossbar device at temperatures of 400 K, 350 K, 300 K, 250 K, 200 K and 150 K. Insert: The four-terminal measurement. **f** The extracted barrier heights of the 0D junction as a function of voltage.

## 0D vdW interfacial ferroelectric diode

We further studied the 0D vdW junction device as a ferroelectric diode. As shown in Fig. 2a, by sweeping the voltage, we found that the rectification can be switched, corresponding to an abrupt resistive switch. Logarithmic plot of the I-V curves is presented in Fig. 2b. Considering that the 0D junction is a homojunction, the reverse of rectification and resistance switch should be caused by a switchable electric polarization of the junction. The rectification and resistive switch are repeatable, showing endurance in multiple tests as is shown in the linear and logarithmic plots. The resistances of the ON and OFF states with a read voltage of 1 V are shown in Fig. 2c, corresponding to an average ON/OFF ratio of 27, which is comparable to PcRAM (10–100)[40], higher than MRAM (2–3)[42], but lower than RRAM (10⁴–10⁵)[41] and NEMS memory (infinite)[39]. The resistance and ON/OFF ratio with other read voltage are presented in Supplementary Fig. 9. The coercive voltage, or writing voltage that leads to the resistive switch is about 4.8 V, as is shown in Fig. 2d, corresponding to a coercive electric field of 3 V/nm. Such a rectification switch and resistive modulation have been reported previously in metal-ferroelectric material-metal trilayer structures known as ferroelectricity diodes, where the sandwiched ferroelectric materials were ferroelectric film materials such as $BiFeO_3$[37], $Pb(Zr_xTi_{1-x})O_3$ (PZT)[50], or $Hf_{1-x}Zr_xO_2$ (HZO)[38]. Here, we demonstrate with the fabricated device that the 0D vdW interface could play as the essential module for ferroelectric diode-like behavior.

Now we discuss the mechanism of the polarization switch at the 0D vdW junction. Although the 0D vdW junction shows a ferroelectric diode-like behavior, the source of the polarization shift should be carefully verified. Two possible mechanisms might contribute to such polarization switch: The first hypothesis, as shown in Fig. 3a, is the electric field induced atomic vacancy transfer across the 0D interface, which was observed at the $MoS_2$-gold interface and generates the resistance switch, as reported in the previous studies[51–54]. The migrating atomic vacancy that carries net charge can cause a shift of the electrical polarization across the 0D interface. The second hypothesis, as shown in Fig. 3b, is that the 0D vdW interface changes its stacking order by atomic scale sliding, which results in interfacial ferroelectricity, reported earlier in subtle 2D vdW interface of hBN, TMDCs, and sandwiched graphene[17–19,21–29]. Given the infeasibility of the in situ observation in this case for the potential single atomic vacancy migration or atomic scale sliding, here we have to design an operable experiment for convincing verification. As shown in the upper insert of Fig. 3c, we spin-coated PMMA of 300 nm thickness to cover the $WS_2$ 0D van der Waals junction device, and repeated the measurement shown in Fig. 2. The conductivity of the 0D interface remained at the same level, however, the phenomenon of rectification switch or resistive modulation disappeared, as shown in Fig. 3c. Since the filling PMMA does not block the migration of the atom vacancy but prevents $WS_2$ nanotube from sliding, the polarization shift should result from the mechanical sliding guaranteed by the superlubricity at the vdW interface. Therefore, the 0D vdW sliding ferroelectric diode device can be regarded as a special branch of NEMS memory[55,56]. A general theory has been proposed and well demonstrated[16,57]. We performed density functional theory (DFT) simulation on the stacked $WS_2$ planes. The interfacial differential charge densities of rhombohedral stacked $WS_2$ planes explicitly show charge redistribution between the top and bottom layers (Fig. 3d, e)[25]. Their line profiles in Fig. 3f show explicit switched electric polarization at the interfaces, and Fig. 3g shows interlayer potential difference of rhombohedral stacked $WS_2$, which is absent in parallel and antiparallel stacked $WS_2$ (Supplementary Fig. 10)[26].

Now we discuss the features of the 0D sliding ferroelectricity for memory:

Firstly, the functioning interface is intrinsically constrained. In Fig. 4a, we compare the scale of such an 0D interface to the single ferroelectric domain of 2D ferroelectric materials or interface. As is seen, the ferroelectricity of 0D interface demonstrated here is at least 178 times smaller than that of 2D ferroelectricity. In fact, the scale of 0D ferroelectricity, which is in the solid state, is nearly comparable to a

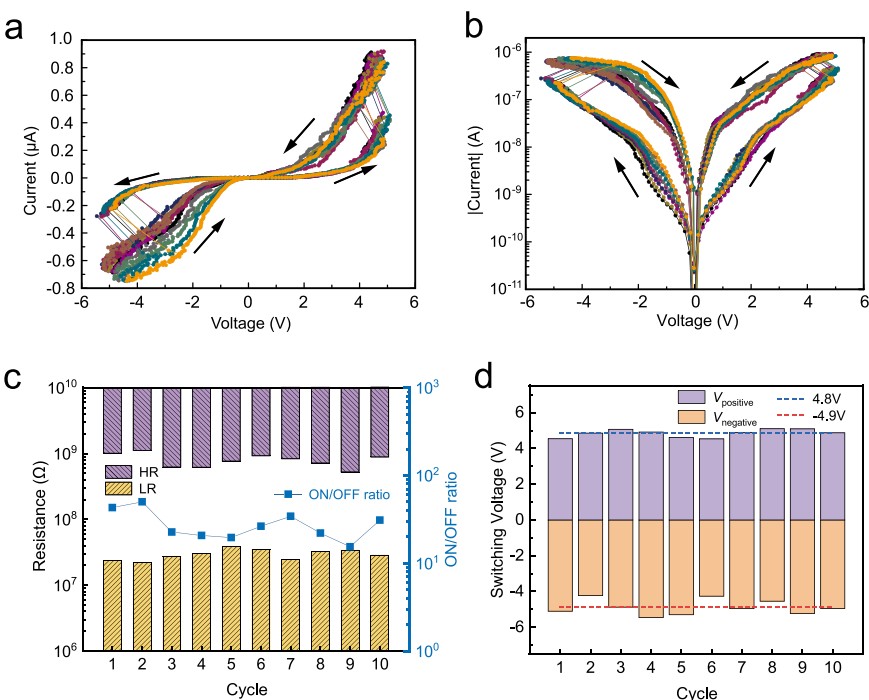

**Fig. 2 | WS₂ nanotube crossbar as the 0D vdW ferroelectric diode. a** I–V characteristic of WS₂ nanotubes crossbar device in the linear coordinate for ten cycles. **b** I–V characteristic in the logarithmic coordinate for ten cycles. **c** The measured high resistance (HR, or OFF-state, by purple bars), low resistance (LR, or ON-state, by yellow bars), and the ON/OFF ratio (blue square) of the device, read at V = 1 V. **d** Switching voltage of the device for 10 cycles and the average value (dotted lines). The measurement is conducted at room temperature unless specified.

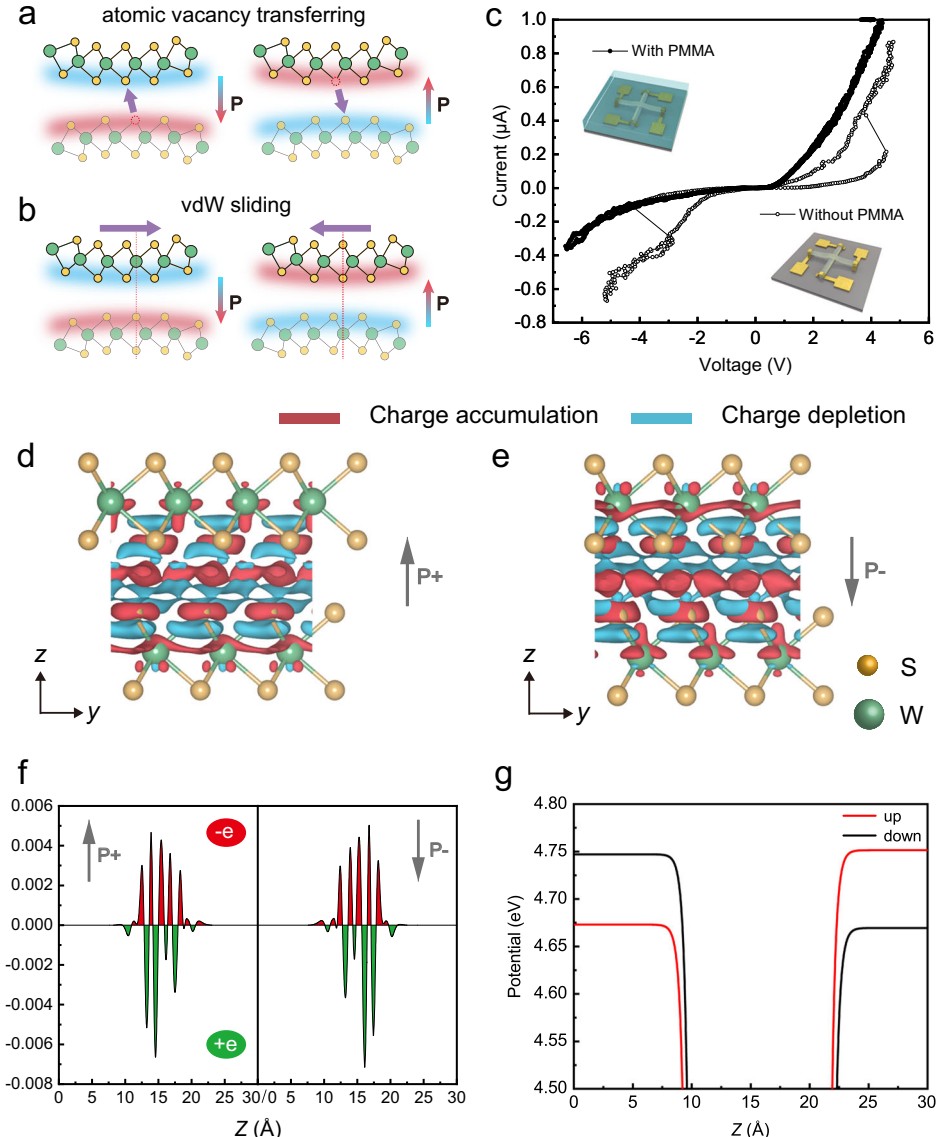

**Fig. 3 | Mechanism and features of the 0D vdW interfacial ferroelectricity.**
**a** Atomic vacancy transferring across $WS_2$ 0D interface. **b** vdW sliding along the $WS_2$ 0D interface. **c** *I–V* curves of the device with and without PMMA. Insert: schematic of the device with and without spin-coated PMMA. **d**, **e** Interlayer differential charge density for the up and down polarizations, respectively. An iso-surface value of $7.0 \times 10^{-5}$ e/bohr³ was used. **f** Their line profiles along *z*, respectively. **g** Interlayer potential.

single molecule electret[31,58]. The core functioning part of the device is only about 5000 atoms large, as shown in Supplementary Fig. 11. Such a constraint provides the possibility for ultra-high density. The constrained area also allows operation at a current of $<10^{-6}$ A, despite the normalized current density of up to $1.15 \times 10^{10}$ A/m². The excessive operation current without an extra restrictor is a major concern of the memory devices such as PcRAM, MRAM, and RRAM, which increases the power consumption of the devices and the local temperature.

Secondly, the coercivity of the 0D sliding ferroelectricity is not hindered by the size effect induced depolarization that limits the scaling down of traditional ferroelectric devices. Instead, the coercive electrical field is even larger than that observed in traditional ferroelectricity, 2D ferroelectricity, or 2D sliding ferroelectricity, as is shown in Fig. 4b. The enhanced coercivity is probably caused by the large pressure dictated by the very small contact area, which requires a higher driving force to initiate the slide and results in the robustness of the polarization. That indicates that the coercion of sliding

ferroelectricity can be tuned by the mechanical condition and opens a gate to tune ferroelectricity with extra mechanical force. The electrical measurement in this work is conducted at room temperature unless specified. We used the thermal simulation to estimate the temperature of 0D ferroelectric interface at work. The highest temperature is about 390 K, as shown in Fig. 4c and Supplementary Fig. 12. Therefore we infer that the Curie temperature of 0D vdW sliding ferroelectricity could be higher than 390 K, which is higher than the Curie temperature obtained by stacked 2D $WSe_2$[23].

Thirdly, the switching speed of the 0D sliding ferroelectric diode is evaluated, which is a key property of performance for memory devices. Programmed electrical pulse is applied to the device with its pulse width experimentally measured, as shown in the Supplementary Fig. 13. The programmed pulse width is 140 ns and the measured pulse width is 160 ns. The resistance switch is observed before and after applying the pulse, as shown in Fig. 4d, indicating that the switch happens within the pulse period of ns.

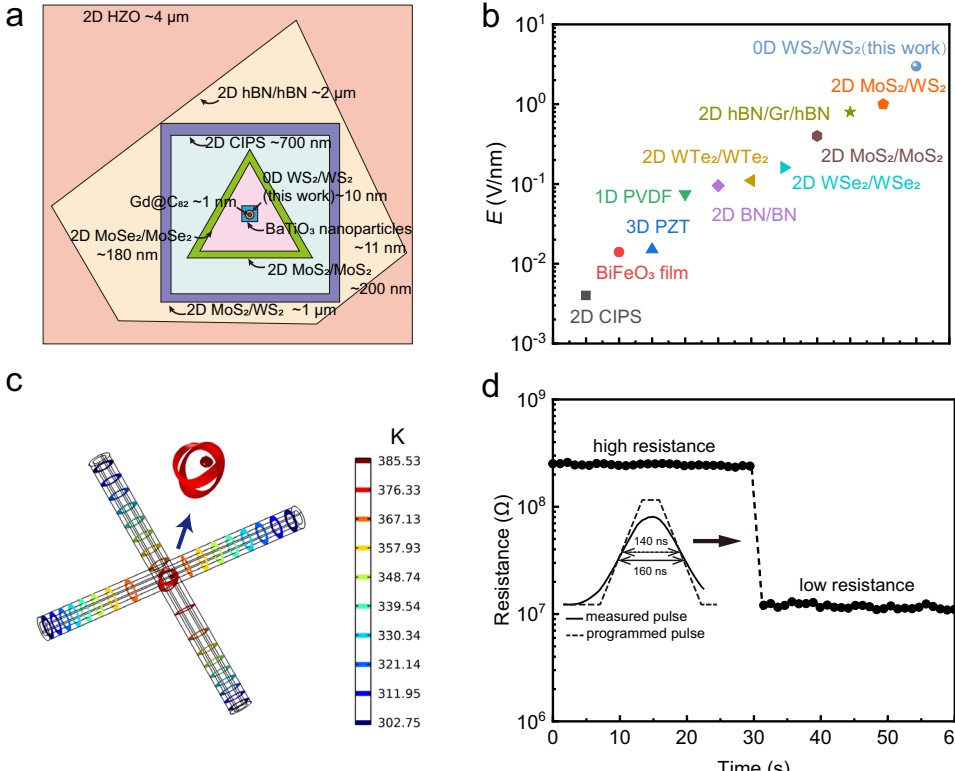

**Fig. 4 | The features of the 0D vdW interfacial ferroelectricity. a** The domains' size of Gd@C$_{82}$[31], 0D WS$_2$/WS$_2$ interface (this work), BaTiO$_3$ nanoparticles[83], 2D MoSe$_2$/MoSe$_2$ interface[26], 2D MoS$_2$/MoS$_2$ interface[27], 2D CuInP$_2$S$_6$ (CIPS)[11], 2D MoS$_2$/WS$_2$ interface[25], 2D hBN/hBN interface[19] and 2D HZO[8]. Non-proportional scale for visualization. **b** Coercive electrical fields of 2D CIPS[84], BiFeO$_3$ film[85], 3D PZT[84], 1D poly(vinylidene fluoride) (PVDF)[84], 2D BN/BN interface[21], 2D WTe$_2$/WTe$_2$ interface[17], 2D WSe$_2$/WSe$_2$ interface[23], 2D MoS$_2$/MoS$_2$ interface[27], 2D hBN/Gr/hBN interface[18], 2D MoS$_2$/WS$_2$ interface[25] and 0D WS$_2$/WS$_2$ interface. **c** Isothermal diagram of the device. (Insert is the diagram around the intersection of nanotubes.) **d** The high/low resistance states before and after the applied pulse. Insert: The programmed and measured pulse waveform.

Fourthly, we investigate the retention and endurance of the 0D sliding ferroelectric diode. The switched ON/OFF states are stable over $5 \times 10^3$ s, as shown in Supplementary Fig. 14, indicating the retention and stabilization of the ferroelectric polarization scaled down to the 0D. However, the endurance of the device versus pulses is less satisfying. Supplementary Fig. 15 shows the cycles of switches of the 0D sliding ferroelectric diode. The device presents tens of switching cycles before the fatigue. The fatigue failure of traditional ferroelectric materials has been long explored, which could be the domain wall pinning, domain nucleation suppression, microcracks, and accumulation of the space or injected charge[59–62]. The endurance of vdW interfacial ferroelectric devices in this work and previous studies is shown in Supplementary Table 1. The failure mechanism of the 0D ferroelectric contact due to fatigue is not clear at this point. It could be possibly explored with the help of the recently developed operando electron microscopy investigation[63].

Lastly, we discuss the major concern of the low yield of the 0D sliding ferroelectric diodes. The major challenge for realization of 0D ferroelectricity, is that it requires rhombohedral stacking[16,26], while at the current stage, chiral control of the WS$_2$ nanotube has not been realized. In this work, we have fabricated 94 devices, among which the rectification switch and resistive modulation are observed in very few (4) devices. For any future application, the chirality-controlled stacking of WS$_2$ nanotube should be further developed towards massive production of the 0D sliding ferroelectric diode. In the very recent studies, the in-situ characterization for the chirality of TMDCs have been developed, and assembly of 1D graphene nanoribbon with chiral control has been demonstrated[64,65]. Despite the low yield, these recent progresses show the potential for addressing this challenge. Note that chirality-controlled incipient growth at the first step and post-growth chirality purification has been achieved in the by far more intensively studied carbon nanotubes[66,67]. These methods could also potentially be extended to the TMDC nanotubes.

## Programmable photovoltaic effect

Besides resistive memory, another important application of ferroelectric diode is its usage as programmable photovoltaics. Traditional ferroelectric materials have a wide bandgap, which limits their availability as visible band photovoltaics. The vdW interfacial ferroelectricity is more flexible in choices of material with suitable bandgap. We have therefore measured the photo response of the 0D ferroelectric diode, as shown in Fig. 5a. The photovoltaic short-circuit current spectrum shows an abrupt edge at around 1.8 eV, as shown in Fig. 5b. Not surprisingly, this photoresponse spectrum is in accordance with the photo absorption spectrum, as shown in Supplementary Fig. 16. The photovoltaic response in a ferroelectric system might originate from various mechanisms, including electric polarization and/or shift current[43,68–70]. The photovoltaic short circuit current versus light power relationship shows a linear to square root transition, which is comparable to the shift current observed in WSe$_2$-black phosphorous heterojunction, as shown in Fig. 5c[68]. Note that the normalization by active area shows a higher photocurrent density for the 0D ferroelectric diode, as shown in Supplementary Fig. 17. The large photocurrent density could be enhanced by the crossbar structure. A hotspot area is formed around the contacting area of the WS$_2$ nanotube junction, as shown in Fig. 5d and Supplementary Fig. 18, which is in accordance with the optical resonance oscillation shown in Supplementary Fig. 19. The polarization of the 0D vdW interface is shifted

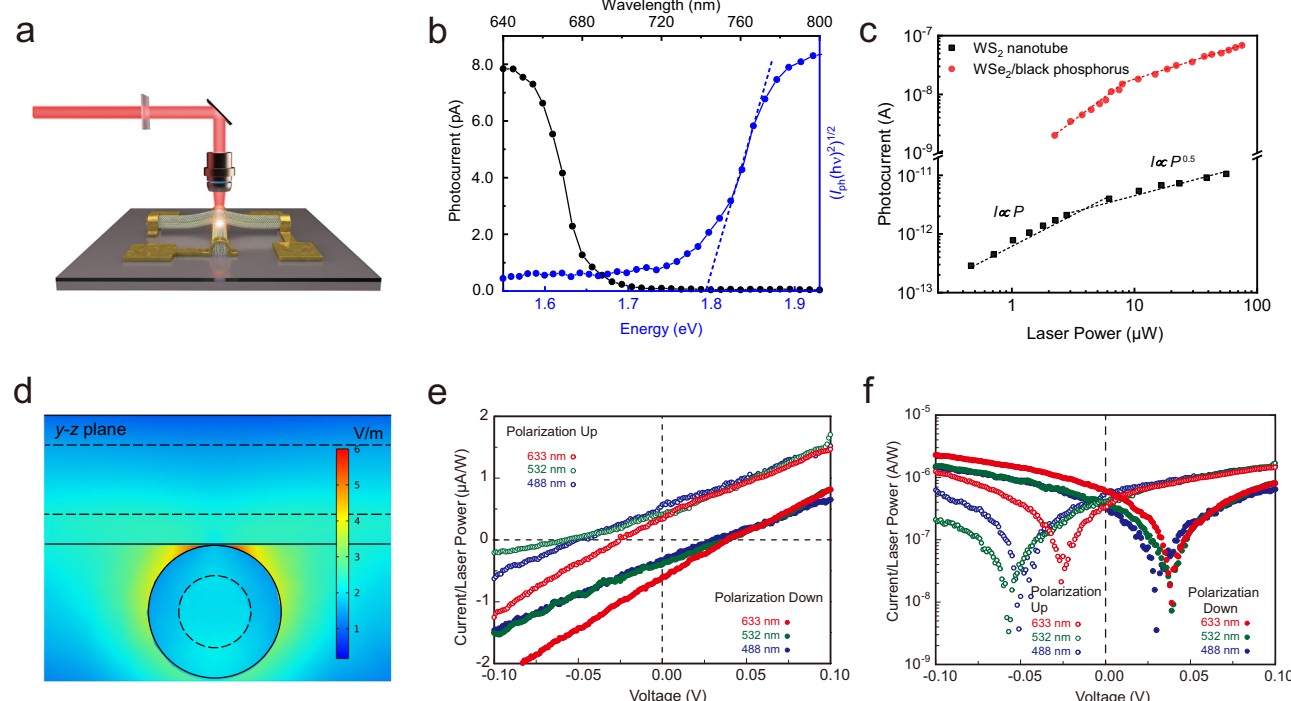

**Fig. 5 | Photovoltaic effect of 0D vdW ferroelectric diode. a** Schematic diagram of the photoresponse measurement. **b** The short circuit photocurrent spectrum of the WS$_2$ nanotubes crossbar with normalized laser power. The photocurrent spectrum indicates a bandgap of about 1.8 eV. **c** Laser power dependence of the photocurrent. In the low-power region, photocurrent is proportional to laser power, whereas the plot shows $I \propto P^{0.5}$ power dependence in the high-power range.

The wavelength of the laser is 645 nm. The results of the WSe$_2$/black phosphorus heterojunction device are extracted from ref. 68. **d** Simulated optical electric field distribution of WS$_2$ nanotubes crossbar illuminated by a laser of 633 nm. **e** *I–V* curves of the device with pre-applied negative/positive voltage, demonstrating a programmable photovoltaic effect. **f** *I–V* curves in logarithmic coordinates.

using a pre-applied voltage of 10 or −10 V as the writing voltage. With the typical laser of 633 nm (red), 532 nm (green), and 488 nm (blue), we demonstrate that the WS$_2$ nanotube junction presents a switchable photovoltaic effect for the visible band light, presented by the *I-V* curves, as shown by the linear plot of Fig. 5e and the logarithmic plot of Fig. 5f. The reversion of the photovoltaic effect is hereby demonstrated with a vdW sliding interface, confirming the spontaneous polarization shift of the 0D vdW interfacial ferroelectric diode.

To summarize, this work demonstrates a 0D ferroelectricity via vdW interfacial sliding that breaks the dimension limit ruled by depolarization. Constrained by the dimension, the 0D vdW interfacial ferroelectricity allows resistance switching operation at a low current level for potential high-density and low-power memory. Moreover, the 0D vdW interfacial ferroelectric diode presents a programmable photovoltaic effect responding to the visible band light. This work provides the strategy to scale ferroelectricity down to 0D, and demonstrates a branch of devices that integrates 0D vdW interfacial ferroelectrics, semiconductor electronics, photovoltaics, and NEMS with potential application to be explored for next-generation versatile electronic systems.

## Methods
### Materials
WS$_2$ nanotubes were obtained using a high temperature chemical reaction between tungsten oxide and H$_2$S gas in a reducing atmosphere. The one-pot self-controlled reaction involved two major steps, i.e., the almost instantaneous growth of tungsten suboxide (W$_{18}$O$_{49}$) nanowhiskers, and the subsequently slow sulfurization of the W$_{18}$O$_{49}$ nanowhiskers into hollow WS$_2$ nanotubes. Oxide nanoparticles consisting of a mixture of different suboxide phases WO$_x$ (with average composition $x = 2.92$) were used as the precursor. For the first step, a series of intermediate reactions occurred, including the reduction of

the precursor into volatile suboxide phase (WO$_{2.75}$); evaporation of WO$_{2.75}$; partial reduction of the vapor into nonvolatile WO$_2$, and condensation of the vapor mixture into stable W$_{18}$O$_{49}$ suboxide phase with a morphology of 1D nanowhiskers. In the second step, the oxide nanowhiskers served as self-consumed template. Here, sulfurization of the W$_{18}$O$_{49}$ nanowhiskers from the surface towards the inner core converted the nanowhiskers into hollow WS$_2$ nanotubes. In the first instant of this reaction, two to three WS$_2$ layers were formed on the surface of the oxide nanowhiskers. Further-on, the conversion of entire oxide nanowhisker into a hollow nanotube was controlled by the rather slow diffusion of the reaction gases and took 3 to 4 hours. The majority of the nanotubes were found to be 2 to 20 μm long with average diameter of about 70 nm, as shown in Supplementary Fig. 2.

### Experimental
The multiwalled WS$_2$ nanotubes are dispersed and stacked onto a silicon substrate with 300 nm silicon oxide using the dry transfer method. The crossbar WS$_2$ nanotubes are positioned via an optical microscope. A layer of PMMA was coated on the surface and heated at 180 °C to evaporate the solvent for 2 min. The pattern of the electrodes was established by electron beam lithography, and Cr/Au = 20/80 nm was deposited by electron beam evaporation as the contact electrodes. Then the resist and the redundant metal are removed by the lift-off process in acetone. The electrical properties were measured by the four-terminal method with a Keithley 4200 semiconductor characterization system or 2400 source measurement meters. AFM was used to characterize the height of the WS$_2$ nanotubes. Opto-electric characterizations were performed with a combination of Raman microscopes, Keithley semiconductor characterization systems, and source meters. Raman microscope equipped with continuous lasers with wavelengths of 633 nm, 532 nm, 488 nm were employed.

## Finite element simulation

Finite element simulation was used to reveal the mechanical deformation, the optical electric field, and temperature distribution in this work. The mechanical properties such as Young's modulus, Poisson's coefficient, the optical properties such as the real and imaginary component of the refractive index, and the thermal properties such as thermal conductivity were obtained from the literature[71–73]. For the mechanical deformation, the displacement was set from 0 to 2 nm with a step size of 0.1 nm, and from 2 to 10 nm with a step size of 1 nm. The maximum strain is automatically given by the results, and the contacting area was extracted by defining the surface with positive normal strain. For the optical simulation, the electromagnetic wave with various polarization direction was set to incident the nanotube crossbar structure from the top direction, the frequency was set from 200 to 1000 THz, corresponding to wavelength of 300–1500 nm. The hotspot of the optic electrical field was shown by the intersection of the structure. The temperature distribution caused by Joule heating was simulated by combining the electrical model and the thermal model, which illustrate the hotspot around the interface.

## DFT simulation

We perform density functional theory calculations using the projector augmented wave method[74], as implemented in the VASP[75,76] package. The kinetic energy cut-off for the plane-wave basis is set to 500 eV, and we treat the exchange-correlation interactions with the Perdew–Burke–Ernzerhof functional[77] in the generalized gradient approximation. A vacuum space exceeding 20 Å is employed to prevent artificial interactions between neighboring image layers. For geometric optimization, the DFT-D3 method[78,79] with Becke–Johnson damping is employed to incorporate the interlayer van der Waals interaction and the structure is fully relaxed until the residual forces on each atom decrease to below 0.01 eV/Å. For electronic self-consistent calculations, we apply a convergence criterion of $10^{-6}$ eV and sample the Brillouin zone using the Monkhorst–Pack scheme with a $12 \times 12 \times 1$ $k$-mesh. The VASPKIT code[80] is utilized for post-processing during the calculations of interlayer potential and differential charge density between layers. A inverted double-bilayer structure[26] is employed to compensate the built-in electric field and ensure the periodic boundary condition in the out-of-plane direction.

## Reporting summary

Further information on research design is available in the Nature Portfolio Reporting Summary linked to this article.

## Data availability

All data are presented in the manuscript or Supplementary Information or available upon reasonable request to the corresponding author.

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

## Acknowledgements

We thank Professor Qing Chen, Professor Lian-Mao Peng, Professor Zhiyong Zhang, Professor Hanchun Wu, and Professor Wende Xiao for help with the experimental, and Ms. Hongyu Zhang, Mr. Rui Liu for useful discussion. R.T. acknowledges the support of the Estate of Manfred Hecht and the Estate of Diane Recanati. He also acknowledges the Perlman Family Foundation. This work is supported by Natural Science Foundation of China (Grant No. 62274011) (Y.G.), (Grant No. 12061131002, Grant No. 12234003) (Y.Y.), Israel Science Foundation (Grant No. 330/16) (A.Z.) and (Grant No. 339/18) (R.T.), China Post-doctoral Science Foundation (Grant No. 2021TQ0043, Grant No. 2021M700437) (L.L.), and Strategic Priority Research Program of Chinese Academy of Sciences (Grant No. XDB30000000) (Y.Y.)

## Author contributions

Conceptualization: Y.G. Methodology: Y.N., Z.Q., H.H.A., Y.G. Material: A.Z, R.T. Simulation: Y.N., L.L. Discussion: Y.N., L.L., Z.Q., H.H.A., X.H., R.T., Y.Y., A.Z., Y.G. Funding acquisition: L.L., A.Z., Y.Y., R.T., Y.G. Visualization: Y.N., Y.G. Supervision: Y.G. Writing—original draft: Y.N. Writing —review and editing: Y.N., L.L., Z.Q., H.H.A., R.T., Y.Y., A.Z., Y.G.

## Competing interests

Planned patent applications by Y.G., Y.N., X.H. (Beijing Institute of Technology), R.T. (Weizmann Institute), and A.Z. (Holon Institute of Technology). The other authors declare no competing interests.
