## [Peer Review File · Nature Communications]

REVIEWER COMMENTS

Reviewer #1 (Remarks to the Author):

Please see attached document.

Reviewer #2 (Remarks to the Author):

The authors demonstrated a 0D ferroelectricity via vdW interfacial sliding at the cross point of two WS₂ nanotubes. Constrained by the dimension, the 0D vdW interfacial ferroelectricity allows resistance switching operation at a low current level for potential high-density and low-power memory, and the ferroelectric diode presents a programmable photovoltaic effect responding to the visible band light. I think this work is interesting and deserves publication, and below are my concerns:

1. The authors may provide more information about the WS₂ nanotubes if available. Are their growth along armchair or zigzag direction? What is the angle at the cross point of two nanotubes? They are multiwall nanotubes, so can the number of layers or walls be estimated?
2. I cannot find the operating temperature for the data in Fig. 2. Can the Curie temperature be estimated?
3. They tuned the pressure in Fig. S4. Can any signal of piezoelectricity be detected?

Reviewer #3 (Remarks to the Author):

In this work, the authors attempted to demonstrate 0D ferroelectricity that may break the dimension limit of depolarization via vdW interfacial sliding.

Although the proposed device structure is interesting and novel, I do not think the present work has reached a level of sufficient maturity for publication. My main concerns are as follows.

1. Among 73 devices fabricated, only 2 devices showed rectification switch and resistance modulation. There is clearly a large uncertainty in the fabrication or in the characterization. Also, the underlying reasons for the low yield are not clearly identified and discussed.
2. The underlying mechanism for the ferroelectricity has not been demonstrated beyond a reasonable doubt. More evidence should be provided to prove the sliding origin via theoretical calculations, for example, DFT methods.
3. The sliding process is not clear. How defects, number of layers, diameters, helicities affect the sliding and the polarization is also not clear.
4. The proposed applications are largely speculative.

Referee Comments

The authors report on “zero-dimensional” (0D) ferroelectricity obtained by atomic sliding at the van der Waals (vdW) interface between two 1D WS₂ nanotubes. Notably, they successfully demonstrate both ferroelectric diode behavior and a programmable photovoltaic effect in a very small device footprint (<10nm x 10nm x 2nm).

This work is timely considering the emergence of (i) interfacial ferroelectric stabilization in low-dimensional vdW materials, and (ii) the ultra-scaled ferroelectric-based memories to realize higher density advanced computing. Therefore, this work is of interest to both the fundamental ferroelectricity community and the applied ferroelectric-based nonvolatile memory community.

While the results are interesting, before I can recommend this manuscript for publication in *Nature Communications*, the authors should address the comments detailed below. In particular, the authors are asked to comment on the qualitative behavior of their ferroelectric diode measurements and provide additional device and structural characterization.

1. [Electrical Measurement] In the ferroelectric diode measurements where the authors are sweeping the voltage, why does the voltage decrease in magnitude during the switching event? Typically, only the current magnitude is expected to change during the switching event, while the voltage magnitude would still increase.

2. [Switching Speed] One other aspect for the realization of dense ferroelectric-based memories is the switching speed of the polarization. As the author's explain, the mechanism of ferroelectricity is due to atomic-scale sliding at the vdW interface. As this is a mechanical effect, this may limit the switching speed. Can authors provide characterization of the switching speed of the ferroelectric diode effect. For example, apply voltages of varying pulse widths and see if the ferroelectric diode effect persists.
3. [Retention] One of the major claims of this work is that the authors are able to overcome depolarization effects, which conventionally prevent ferroelectric stabilization down to the ultra-small/thin regime. To provide more validity to this claim, the authors are asked to perform retention characterization of their ferroelectric diode devices. How stable are the two states over time?
4. [Structural Characterization] The authors have not provided any structural characterization showing rhombohedral stacking in their working devices. The authors should provide some evidence (e.g. from x-ray or optical spectroscopy, as spectroscopic techniques are not as sensitive to the volume of material) for inversion symmetry breaking and/or rhombohedral stacking. Also, the authors include some references to support their claim of the requirement of rhombohedral stacking to realize 0D ferroelectricity (Lines 169-171).

5. [Minor: Methods] (A) The authors mention they used finite-element methods to help determine the contacting area of the interface (Line 77). The authors should include a description of their methodology either in their Method's section or Supplementary Information. (B) Similar information about the optical electric field distribution simulation shown in Fig. 4d should also be provided.
6. [Minor: Write Voltage] For the programmable photovoltaic effect experiment, the authors apply ± 10 V to set the polarization of the device. What is the reason for applying such a large voltage to set the polarization, when in their ferroelectric diode measurements, ± 6 V was enough to switch the polarization? On another note, does the ON/OFF ratio improve when using the higher write voltage?
7. [Minor: Relevant Literature]
 - (A) The authors did a good job of citing the key work on vdW stacked ferroelectrics, but they should cite a few more key references on ultrathin ferroelectricity (Lines 27-30), listed below:
 - [one of the 1st demonstrations of atomic-scale thickness ferroelectricity in a 2D material] Chang, K. *et al.* Discovery of robust in-plane ferroelectricity in atomic-thick SnTe. *Science* **353**, 274–278 (2016).
 - [example of atomic-scale thickness ferroelectricity in a 3D material] Cheema *et al.* Emergent ferroelectricity in subnanometer binary oxide films on silicon. *Science* **376**, 648–652 (2022).
 - (B) The authors should also cite the use of vdW ferroelectrics for memory applications, e.g.:
 - Wang, X. *et al.* Van der Waals engineering of ferroelectric heterostructures for long-retention memory. *Nat. Commun.* **12**, 1109 (2021).
 - Wu, J. *et al.* High tunnelling electroresistance in a ferroelectric van der Waals heterojunction via giant barrier height modulation. *Nat. Electron.* **3**, 466–472 (2020).
 - (C) The authors should cite more work on vdW ferroelectrics displaying the photovoltaic effect:
 - Li, Y. *et al.* Enhanced bulk photovoltaic effect in two-dimensional ferroelectric CuInP2S6. *Nat. Commun.* **12**, 5896 (2021).
 - Yang, D. *et al.* Spontaneous-polarization-induced photovoltaic effect in rhombohedrally stacked MoS2. *Nat. Photonics* **16**, 469–474 (2022).

Reviewer 1

The authors report on “zero-dimensional” (0D) ferroelectricity obtained by atomic sliding at the van der Waals (vdW) interface between two 1D WS₂ nanotubes. Notably, they successfully demonstrate both ferroelectric diode behavior and a programmable photovoltaic effect in a very small device footprint (<10nm x 10nm x 2nm).

This work is timely considering the emergence of (i) interfacial ferroelectric stabilization in low-dimensional vdW materials, and (ii) the ultra-scaled ferroelectric-based memories to realize higher density advanced computing. Therefore, this work is of interest to both the fundamental ferroelectricity community and the applied ferroelectric-based nonvolatile memory community.

We are grateful that with the full comprehension of the recent progresses in the field of interfacial ferroelectricity and ferroelectric-based nonvolatile memory, Reviewer 1 provided us not only the highly positive comment, but also very specific guidance on how to further improve the quality of the manuscript.

While the results are interesting, before I can recommend this manuscript for publication in Nature Communications, the authors should address the comments detailed below. In particular, the authors are asked to comment on the qualitative behavior of their ferroelectric diode measurements and provide additional device and structural characterization.

The questions from Reviewer 1 are very informative and useful. Our point-to-point responses are as below.

1. [Electrical Measurement] In the ferroelectric diode measurements where the authors are sweeping the voltage, why does the voltage decrease in magnitude during the switching event? Typically, only the current magnitude is expected to change during the switching event, while the voltage magnitude would still increase.

We thank Reviewer 1 for proposing this question. Fig. 2a and 2b show the decreased magnitude of voltage during the switching event. The decreased magnitude of magnitude is determined by the method of four-terminal measurements (also known as the four-probe method, Fig. R1a) applied in this work.

As shown in the insert of Fig. 1e and Fig. R1b, the crossed nanotubes form four extended arms, i. e. Nanotube1-Arm1, Nanotube1-Arm2, Nanotube2-Arm3, Nanotube2-Arm4. The sweeping voltage, set up with the varying magnitude in steps, is applied between Arm 1 and Arm 3. The current goes along Arm 1, the 0D van der Waals interface, and Arm 3. We have $current = V_{set} / (R_{Arm1} + R_{Arm3} + r)$, where R_{Arm} is the resistance of the arm, including the metal-nanotube contact resistance and the resistance of the nanotube. r is the interfacial resistance, and V_{set} is the set voltage. During the switching event, the resistance r decreases, the *current* increases. The voltage drop across the interface, collected as the difference of voltages from Arm 2 and Arm 4, $voltage = current \cdot r = V_{set} \cdot r / (R_{Arm1} + R_{Arm3} + r)$, decreases. The four-terminal measurement adopted in this work shows the intrinsic change of the interfacial resistance by eliminating the effect of the resistance of the nanotubes and electrical contacts.

For the concern that “Typically, only the current magnitude is expected to change during the

switching event, while the voltage magnitude would still increase.” This is usually seen in the plotted results from two-terminal measurement, if the plot uses the set-up value of the sweeping voltage. In other word, the value of the sweeping voltage is programmed to increase or decrease. It should be noted that if the current reaches the compliance during the switch event (which often happens for RRAM, Pc-RAM, et. al.), the voltage amplitude actually applied should be determined by the compliant current and cannot be reflected by the set-up value of the sweeping voltage.

Fig. R1 | Equivalent circuit of the four-terminal measurement. a, Schematic diagram of the four-terminal measurement. **b,** Schematic circuit diagram of the four-terminal measurement applied in this work.

We have supplemented the above discussion in the revised version of the manuscript (line 96-97) and supplementary information (Supplementary Fig. 7, line 36-39).

“We used the four-terminal measurement to characterize the electrical performance of the 0D interface, as shown in the insert of Fig. 1e and Supplementary Fig. 7.”

2. [Switching Speed] One other aspect for the realization of dense ferroelectric-based memories is the switching speed of the polarization. As the author’s explain, the mechanism of ferroelectricity is due to atomic-scale sliding at the vDW interface. As this is a mechanical effect, this may limit the switching speed. Can authors provide characterization of the switching speed of the ferroelectric diode effect. For example, apply voltages of varying pulse widths and see if the ferroelectric diode effect persists.

We thank Reviewer 1 for the constructive suggestion. The switching speed is a key performance of memory devices. The polarization switching speed of the interfacial sliding ferroelectricity has not been reported yet in our nearest knowledge. It will be of great interest to investigate the switching speed for the first time.

As shown in Fig. R2a, the pulse set with a rise time of 100 ns, fall time of 100 ns, and pulse width of 140 ns was measured and applied to the device. The measured waveform of the pulse shows an actual full width at half maximum of 160 ns. Fig. R2b exhibits the resistive switch before and after the writing voltage pulse. After the pulse, the device changes from a high resistance state to a low resistance state, corresponding to a switching time of 160 ns. It is worth noting that the pulse width adopted in this work is limited by the equipment available in our lab (4200 semiconductor characterization system with a pulse generator), and the intrinsic switching time could be smaller. In recent research, the molecular dynamic simulation shows that the status relaxation from polar to non-polar states of the sliding ferroelectricity can be within 20 ps [Advanced Functional Materials, 2023: 2301105].

Fig. R2 | The switching speed of the device. a, Programmed and measured waveforms of the electrical pulse. **b,** The high/low resistance states before and after the applied pulse.

We have supplemented the above discussion in the revised version of the manuscript (Figure 4d, line 190-196) and the supplementary information (Supplementary Fig. 13, line 59-61).

“Thirdly, the switching speed of the 0D sliding ferroelectric diode is evaluated, which is a key property of performance for memory devices. Programmed electrical pulse is applied to the device with its pulse width experimentally measured, as shown in Supplementary Fig. 13. The programmed pulse width is 140 ns and the measured pulse width is 160 ns, which is the shortest pulse available from our lab. The resistance switch is observed before and after applying the pulse, as shown in Fig. 4d, indicating that the switch happens within the pulse period of ns.”

3. [Retention] One of the major claims of this work is that the authors are able to overcome depolarization effects, which conventionally prevent ferroelectric stabilization down to the ultra-small/thin regime. To provide more validity to this claim, the authors are asked to perform retention characterization of their ferroelectric diode devices. How stable are the two states over time?

We thank Reviewer 1 for the constructive comment. Previous research has demonstrated stable sliding ferroelectricity in 2D van der Waals assemblies [Science 372, 1458–1462 (2021). Nat. Nanotechnol. 17, 367–371 (2022). Science 376, 973–978 (2022)]. We understand that Reviewer 1’s concern is if the scaling down to 0D causes the loss of stabilization, which is fundamental to the nanoscale ferroelectricity. Fig. R3 shows the retention of the low and high resistance status of the device measured at room temperature, corresponding to the ON and OFF state of the memroy. The ON/OFF state maintained well distinguishable after more than 1.5h, showing retention of the polarization.

Fig. R3 | Retention characteristics of the ON (black) and OFF (red) states.

We have supplemented the above discussion in the revised version of the manuscript (line 196-198) and the supplementary information (Supplementary Fig. 14, line 62-64).

“The switched ON/OFF states are stable over 5×10^3 s, as shown in Supplementary Fig. 14, indicating the retention and stabilization of the ferroelectric polarization scaled down to the 0D.”

4. [Structural Characterization] The authors have not provided any structural characterization showing rhombohedral stacking in their working devices. The authors should provide some evidence (e. g. from x-ray or optical spectroscopy, as spectroscopic techniques are not as sensitive to the volume of material) for inversion symmetry breaking and/or rhombohedral stacking. Also, the authors include some references to support their claim of the requirement of rhombohedral stacking to realize 0D ferroelectricity (Lines 169-171).

We thank Reviewer 1 for the comment. We agree that the stacking order is critical to the generation of van der Waals interfacial ferroelectricity. For example, the previous work [Nat. Nanotechnol. 17, 367–371 (2022)] has proven that the sliding ferroelectricity exist only with the rhombohedral stacking, while polarization is absent for parallel stacking and anti-parallel stacking.

The x-ray diffraction of the WS₂ nanotubes is shown in Fig. R4a. The peak of the (002) plane of the nanotubes shows an interlayer distance of 6.4 Å. However, the signal of the x-ray diffraction is from multiple nanotubes, rather than the contacting 0D interface, therefore we were not able to obtain the stacking order of the 0D interface with the method of x-ray diffraction.

Raman spectroscopy has been used to characterize the diameter and chirality of single-walled carbon nanotubes [Accounts of Chemical Research, 2002, 35(12): 1070-1078.]. Fig. R4b shows the typical Raman spectrum from a single multiwall WS₂ nanotube adopted in this work. As shown in Fig. R4b, the E_{2g}¹ mode involves an in-phase vibration of the W atoms with respect to the S atoms vibrating in the opposite direction in-phase, while the A_{1g} mode arises from the S atoms moving in in-phase and in out-of-plane directions. However, still, we could not find an effective method to determine the chirality of the outmost layer of the multiwall WS₂ nanotube.

We do notice that the chirality of the few layer WS₂ nanotube can be characterized with high-resolution transmission electron microscopy, as shown by Maya Bar Sadan et. al [Proceedings of

the National Academy of Sciences, 2008, 105(41): 15643-15648.]. However, we are not able to adopt this technology in this work because the WS₂ nanotubes used in this work are with more layers so that the length of the nanotube can facilitate the assembly. Recently, stacking of 1D nanoribbons was demonstrated with the twist angle well controlled, which requires delicate STM manipulation. [Nat. Commun. 14, 1018 (2023)]. Another recent work demonstrates a conceptually new type of scanning probe microscope—the quantum twisting microscope—capable of performing local chirality with high angular and positional precision. The quantum twisting microscope opens the way for creating highly controllable new interfaces between a large variety of quantum materials [Nature, 2023, 614(7949): 682-687].

Fig. R4 | Structural Characterization of WS₂ nanotube. a, x-ray spectrum and **b**, Raman spectrum of the WS₂ nanotubes.

We have supplemented the above discussion in the revised version of the supplementary information (Supplementary Fig. 3, line 15-21)

“**Supplementary Fig. 3 | Structural Characterization of WS₂ nanotube. a**, The x-ray diffraction of the WS₂ nanotubes. The peak of the (002) plane of the nanotubes shows an interlayer distance of 6.4 Å. **b**, The Raman spectrum of the WS₂ nanotube. The E'₂g mode involves an in-phase vibration of the W atoms with respect to the S atoms vibrating in the opposite direction in-phase, while the A₁g mode arises from the S atoms moving in in-phase and in out-of-plane directions.”

We have also added reference [Nat. Nanotechnol. 17, 367–371 (2022) ACS. Nano. 2017, 11, 6, 6382–6388], and the results of density functional theory simulation in Fig. 3, to support the requirement of rhombohedral stacking to realize 0D ferroelectricity. (manuscript, line 200 to 202)

“The major challenge for realization of 0D ferroelectricity, is that it requires rhombohedral stacking^{16,26}, while at the current stage, chiral control of the WS₂ nanotube has not been realized.”

5. [Minor: Methods] (A) The authors mention they used finite-element methods to help determine the contacting area of the interface (Line 77). The authors should include a description of their methodology either in their Method's section or Supplementary Information. (B) Similar information about the optical electric field distribution simulation shown in Fig. 4d should also be provided.

We thank Reviewer for the kind reminder. In this work, we used finite-element method to determine the contacting area, to estimate the temperature distribution, and to illustrate the hotspot of the optical electric field. We supplement information about the finite element simulation as follows:

Finite element simulation was used to reveal the mechanical deformation, the optical electric field, and temperature distribution in this work. The mechanical properties such as Young's modulus, Poisson's coefficient, the optical properties such as the real and imaginary component of the refractive index, and the thermal properties such as thermal conductivity were obtained from the literature [reference 66, 67, and 68]. For the mechanical deformation, the displacement was set from 0 to 2 nm with the step size of 0.1 nm, and from 2 to 10 nm with the step size of 1 nm. The maximum strain is automatically extracted from the results, and the contacting area was determined by defining the area of the surface with positive normal strain. For the optical simulation, the electromagnetic wave with various polarization direction was set to incident the nanotube crossbar structure from the top direction, the frequency was set from 200-1000 THz, corresponding to the wavelength of 300 nm to 1500 nm. The hotspot of the optic electrical field was shown by the intersection of the structure. The temperature distribution caused by Joule heating was simulated by combining the electrical model and the thermal model, which illustrate the hotspot around the interface.

We have added the above content to the revised manuscript (line 276 to 291).

6. [Minor: Write Voltage] For the programmable photovoltaic effect experiment, the authors apply ± 10 V to set the polarization of the device. What is the reason for applying such a large voltage to set the polarization, when in their ferroelectric diode measurements, ± 6 V was enough to switch the polarization? On another note, does the ON/OFF ratio improve when using the higher write voltage?

We thank Reviewer 1 for the question. The voltage applied for the programmable photovoltaic effect experiment is different from the voltage collected using the four-terminal method, as explained in the above response. The voltage applied is larger than the measured voltage drop on the 0D interface. We did not use higher voltage for the writing here to avoid potential oxidation to the nanotube and thermal failure of the device.

We have addressed in the manuscript to avoid misunderstanding about the applied voltage. (line 230-line 231)

“The polarization of the 0D vdW interface is shifted using a pre-applied voltage of 10 V or -10 V as the writing voltage.”

7. [Minor: Relevant Literature]

(A) The authors did a good job of citing the key work on vdW stacked ferroelectrics, but they should cite a few more key references on ultrathin ferroelectricity (Lines 27-30), listed below:

- [one of the 1st demonstrations of atomic-scale thickness ferroelectricity in a 2D material] Chang, K. et al. Discovery of robust in-plane ferroelectricity in atomic-thick SnTe. *Science* 353, 274–278 (2016).

- [example of atomic-scale thickness ferroelectricity in a 3D material] Cheema et al. Emergent ferroelectricity in subnanometer binary oxide films on silicon. *Science* 376, 648–652 (2022).

(B) The authors should also cite the use of vdW ferroelectrics for memory applications, e.g.:

- Wang, X. et al. Van der Waals engineering of ferroelectric heterostructures for long-retention memory. *Nat. Commun.* 12, 1109 (2021).

- Wu, J. et al. High tunnelling electroresistance in a ferroelectric van der Waals heterojunction via giant barrier height modulation. *Nat. Electron.* 3, 466–472 (2020).

(C) The authors should cite more work on vdW ferroelectrics displaying the photovoltaic effect:

- Li, Y. et al. Enhanced bulk photovoltaic effect in two-dimensional ferroelectric CuInP₂S₆. *Nat. Commun.* 12, 5896 (2021).
- Yang, D. et al. Spontaneous-polarization-induced photovoltaic effect in rhombohedrally stacked MoS₂. *Nat. Photonics* 16, 469–474 (2022).

We thank Reviewer 1 very much for the suggestion, the references listed are important and we have added them in the revised manuscript:

“For example, the recent research on two-dimensional (2D) ferroelectricity has brought the thickness down to the atomic limit, including the ultrathin amorphous doped hafnium/zirconium oxide⁸⁻¹⁰, the monolayer ferroelectric materials¹¹⁻¹⁵, and the subtle van der Waals (vdW) stacked assembly¹⁶⁻³⁰.”

[15 Chang, K. et al. Discovery of robust in-plane ferroelectricity in atomic-thick SnTe. *Science* 353, 274-278 (2016).]

[10 Cheema, S. S. et al. Emergent ferroelectricity in subnanometer binary oxide films on silicon. *Science* 376, 648-652 (2022).]

“Ferroelectrics are important candidates for nonvolatile memory as part of the quest for denser storage, lower power consumption, and neuromorphic computing³²⁻³⁶.”

[36 Wang, X. et al. Van der Waals engineering of ferroelectric heterostructures for long-retention memory. *Nature communications* 12, 1109 (2021).]

[35 Wu, J. et al. High tunnelling electroresistance in a ferroelectric van der Waals heterojunction via giant barrier height modulation. *Nature Electronics* 3, 466-472, doi:10.1038/s41928-020-0441-9 (2020).]

The photovoltaic response in a ferroelectric system might originate from various mechanisms, including electric polarization and/or shift current^{43,63-65}.

[65 Li, Y. et al. Enhanced bulk photovoltaic effect in two-dimensional ferroelectric CuInP₂S₆. *Nature communications* 12, 5896 (2021).]

[64 Yang, D. et al. Spontaneous-polarization-induced photovoltaic effect in rhombohedrally stacked MoS₂. *Nature Photonics* 16, 469-474 (2022).]

Reviewer 2

The authors demonstrated a 0D ferroelectricity via vdW interfacial sliding at the cross point of two WS₂ nanotubes. Constrained by the dimension, the 0D vdW interfacial ferroelectricity allows resistance switching operation at a low current level for potential high-density and low-power memory, and the ferroelectric diode presents a programmable photovoltaic effect responding to the visible band light. I think this work is interesting and deserves publication, and below are my concerns:

We thank Reviewer 2 for the positive comment to this work. We appreciate Reviewer 2's effort to help us to further improve the quality of this manuscript. We have supplemented results and discussion with our best effort in the revised version of the manuscript. The point-to-point responses are as follows:

1. The authors may provide more information about the WS₂ nanotubes if available. Are their growth along armchair or zigzag direction? What is the angle at the cross point of two nanotubes? They are multiwall nanotubes, so can the number of layers or walls be estimated?

We thank Reviewer 2 for the valuable comment. We have supplemented the results of the transmission electron microscopy, the x-ray diffraction, and the Raman spectroscopy in the revised version of the manuscript, as shown in Fig. R5. From the results of transmission electron microscopy, it can be seen that the growth of the multiwall WS₂ nanotubes used in this research is along random direction. We notice that the chirality of the very thin WS₂ nanotube can be characterized with high-resolution transmission electron microscopy, as shown by Maya Bar Sadan et. al. [Proceedings of the National Academy of Sciences, 2008, 105(41): 15643-15648.] However, we are not able to adopt this technology in this work because the WS₂ nanotubes used in this work are with more layers so that the length of the nanotube can facilitate the assembly. Recently, stacking of 1D nanoribbons was demonstrated with the twist angle well controlled, which requires delicate STM manipulation. [Nat. Commun. 14, 1018 (2023)]. Another recent work demonstrates a conceptually new type of scanning probe microscope—the quantum twisting microscope—capable of performing local chirality with high angular and positional precision. The quantum twisting microscope opens the way for creating highly controllable new interfaces between a large variety of quantum materials [Nature, 2023, 614(7949): 682-687]. The number of layers of the nanotube shown in Fig. R5a and Fig. S2b is 11. The number of layers might vary among different nanotubes, which ranges from 8 to 35.

Fig. R5 | Structure characterization of WS₂ nanotube. **a**, Transmission electron microscopy image of a multiwall WS₂ nanotube. **b**, Electron diffraction pattern of a multiwall WS₂ nanotube. **c**, x-ray spectrum and **d**, Raman spectrum of the WS₂ nanotubes we used.

2. I cannot find the operating temperature for the data in Fig. 2. Can the Curie temperature be estimated?

We thank Reviewer 2 for the valuable comment. The results of electrical characterization in Fig. 2 were obtained from measurement at room temperature.

The Curie temperature of stacked bilayer WSe₂ in a ferroelectric field-effect transistor device was reported to be 353K [Nano Letters, 2022, 22(3): 1265-1269]. For the ferroelectric diode devices in this research, the writing current generates Joule heat, which makes the temperature of the core region hard to estimate. With finite element thermal simulation, we obtain the temperature distribution of the stacked WS₂ nanotube device. As shown in Fig. R6, the highest temperature appears around the junction area, which reaches up to about 390 K. Considering that the device still shows the polarization switch, therefore it can be inferred that the Curie temperature should be higher than 390 K. The polarization could be enhanced by the concentrated stress of the constrained interface. We did not conduct high temperature measurement to prevent the potential oxidation and failure to the nanotubes.

Fig. R6 | Simulated temperature distribution of the crossed WS₂ nanotubes. **a**, The temperature distribution, maximum and minimum temperature of the device. **b**, Isotherm distribution diagram of the device. Insert: an enlarged temperature distribution diagram around the intersection.

We have added the above content to the revised manuscript (Fig. 4c line 184 to 189) and the supplementary information (Supplementary Fig. 12, line 57-58).

“The electrical measurement in this work is conducted at room temperature unless specified. we used the thermal simulation to estimate the temperature of 0D ferroelectric interface at work. The highest temperature is about 390 K, as shown in Fig. 4c and Supplementary Fig. 12. Therefore we infer that the Curie temperature of 0D vdW sliding ferroelectricity could be higher than 390 K, which is higher than the Curie temperature obtained by stacked 2D WSe₂²³.”

3. They tuned the pressure in Fig. S4. Can any signal of piezoelectricity be detected?

We thank Reviewer 2 for the valuable comment. In previous works, the signal of piezoelectricity was collected using piezo-response force microscopy. However, we were not able to conduct the piezoelectricity measurement because 1) unlike the 2D vdW assembly that can expose the electric potential to the scanning probe microscopy, the 0D interface is secluded and the probe cannot collect signal directly, and 2) the amount of charge generated due to piezoelectricity is very small. The functioning area of the 0D interface is less than 100 nm². The estimated piezoelectric charge is 3×10^{-19} C per cycling of the pressure, which is less than 2 electrons. Such a small amount of charge is hard to be directly detected. Even the pressure can be tuned with the frequency of up to 1 kHz, the estimated alternating current is less than 3×10^{-16} A, such a small current is far below the detection limit of the equipment and would merge by noise or signal of triboelectrification that accompanies the application of deformation.

Reviewer 3

In this work, the authors attempted to demonstrate 0D ferroelectricity that may break the dimension limit of depolarization via vdW interfacial sliding.

Although the proposed device structure is interesting and novel, I do not think the present work has reached a level of sufficient maturity for publication. My main concerns are as follows.

We thank Reviewer 3 for the comment of the manuscript. By reading the comment, we understand that Reviewer 3's concern is about the yield, the mechanism to be verified by DFT, and the potential application of the 0D sliding ferroelectricity. Following the suggestions by Reviewer 3, we analyze the reasons for low yield and discuss potential solutions to this problem, conduct the DFT simulation, and discuss how sliding ferroelectric diode can be applied, for example, in neuromorphic computing at the edge of sensors. We would also like to address the scientific progresses presented in this work, i. e. the construction of interfacial ferroelectricity ultimately scaled down to 0D, the ferroelectric diode device by van der Waals sliding, and the programmable photovoltaics by sliding ferroelectricity. Our point-to-point responses to the concerns are as below.

1. Among 73 devices fabricated, only 2 devices showed rectification switch and resistance modulation. There is clearly a large uncertainty in the fabrication or in the characterization. Also, the underlying reasons for the low yield are not clearly identified and discussed.

We thank Reviewer 3 for the comment. The low yield of the device is indeed a major concern for the massive production and large-scale integration of the 0D ferroelectric diode devices demonstrated by this work. The fundamental reason underlying for the low yield could be that the realization of sliding ferroelectricity requires alignment of rhombohedral stacking. This proposes critical challenge to the chiral controlment during the fabrication process, including the controlled growth with designed chirality and the fast characterization of chirality of the nanotube.

These two major challenges have been addressed to a certain degree by decades of intensive studies for carbon nanotubes, which also present the chirality dependent physical properties. Transistor based carbon nanotube were first demonstrated back in 1990s [Nature, 1998, 393(6680): 49-52], and has been regarded as an important potential candidate to extend the Moore's law. However, the fatal problem is that the chirality of carbon nanotubes results in differences of metallic state or semiconductor state. In 2014, Feng Yang et al. showed that carbon nanotubes of uniform chirality can be massively produced directly with the purity of higher than 92% [Nature, 2014, 510(7506): 522-524.]. In 2020, by combining multiple-dispersion sorting and dimension-limited self-alignment methods, well-aligned (within 9° of alignment), high-purity (better than 99.9999%), and high-density (tunable between 100 and 200 carbon nanotubes /mm) arrays of carbon nanotubes was firstly prepared on 4-inch silicon wafers [Science, 2020, 368(6493): 850-856]. The chirality of the carbon nanotube can be characterized with transmission electron microscopy or Raman spectroscopy [Accounts of Chemical Research, 2002, 35(12): 1070-1078.].

For the WS₂ nanotube used in this work, we do acknowledge that the major challenges of chiral control and chiral characterization of WS₂ nanotubes remains in suspense. Some progress has been made in the chiral characterization of nanotubes at the microscopic level, for example, Maya Bar Sadan used high-resolution transmission electron microscopy to reveal the chirality of the different shells and provided a better understanding of few layered WS₂ nanotube growth mechanism

[Proceedings of the National Academy of Sciences, 2008, 105(41): 15643-15648]. However, we are not able to adopt this technology in this work because the WS₂ nanotubes used in this work are with more layers so that the length of the nanotube can facilitate the assembly. A very recent work has demonstrated a conceptually new type of scanning probe microscope—the quantum twisting microscope—that can perform local chirality characterization of TMDC. [Nature, 2023, 614(7949): 682-687]. Another recent work used scanning tunneling microscope for the manipulation of flat graphene nanoribbon [Nat. Commun. 14, 1018 (2023)]. We conducted scanning tunneling microscopy on the WS₂ nanotube sample, however, it did not achieve chiral characterization probably due to the obscure from the curvature surface of the nanotube. Still, the recent progress does show that there is potential solution to the challenge of chiral characterization and, possibly, chiral control, despite the low efficiency at the current stage. This progress is not the topic of this research.

In coexist of the current challenges, the presented progress in this study, however, should not be neglected, just as the significance of the demonstrated carbon nanotube transistor should not be dismissed. In this study, we realize the solid-state 0D ferroelectricity with the sliding van der Waals interface, the fabricated device of 0D van der Waals interfacial ferroelectricity shows not only the electrical resistance switch, but also the programmable photovoltaic effect that covers the nearly full visible range. We believe the progress present in this work is of interest to not only the fundamental ferroelectricity community, but also the applied ferroelectric-based nonvolatile memory community.

We have summarized the above discussion to the revised manuscript (line199 - line 211):

“Lastly, we discuss the major concern of the low yield of the 0D sliding ferroelectric diodes. The major challenge for realization of 0D ferroelectricity, is that it requires rhombohedral stacking, while at the current stage, chiral control of the WS₂ nanotube has not been realized. In this work, we have fabricated 94 devices, among which the rectification switch and resistive modulation are observed in very few (3) devices. Still, the chirality-controlled stacking of WS₂ nanotube is to be developed for the massive production of the 0D sliding ferroelectric diode. In the very recent studies, the in-situ characterization for the chirality of TMDCs have been developed, and assembly of 1D graphene nanoribbon with chiral control has been demonstrated^{59,60}. Despite the low yield, these recent progresses show the potential resolution to this challenge. By far the intensively studied carbon nanotubes have demonstrated chirality-controlled incipient growth at the first step and post-growth chirality purification^{61,62}. These methods could also potentially be extended to the TMDC nanotubes.”

2. The underlying mechanism for the ferroelectricity has not been demonstrated beyond a reasonable doubt. More evidence should be provided to prove the sliding origin via theoretical calculations, for example, DFT methods.

We thank Reviewer 3 for the useful suggestion. We performed density functional theory (DFT) simulation on the stacked WS₂ interface, as shown in Fig. R7. The interfacial differential charge densities of rhombohedral stacked WS₂ explicitly show charge redistribution between the top and bottom layers. Their line profiles show explicit switched electric polarization at the interfaces, and Fig. R7d shows a potential difference of 78.4 meV and -77.5 meV when the polarization is up and down, respectively. We also note that the underlying mechanism for the sliding ferroelectric has

been generalized, which support the mechanism of the sliding ferroelectricity in a well-constructed theory frame [Physical Review Letters 130, 146801 (2023)].

Fig. R7 | Charge density and polarization. **a, b**, Interlayer differential charge density for the up and down polarizations, respectively. An iso-surface value of 7.0×10^{-5} e/bohr³ was used. **c**, The line profiles along z , respectively. **d**, Interlayer potential.

We have included the above discussion to the revised manuscript (line155-line 162, Fig. 3d-g):

“The general theory has been proposed and well demonstrated^{16,57}. We performed density functional theory (DFT) simulation on the stacked WS₂. The interfacial differential charge densities of rhombohedral stacked WS₂ explicitly show charge redistribution between the top and bottom layers (Fig. 3d, e)²⁵. Their line profiles in Fig. 3f show explicit switched electric polarization at the interfaces, and Fig. 3g shows interlayer potential difference of rhombohedral stacked WS₂, which is absent in parallel and antiparallel stacked WS₂ (Supplementary Fig. 10)²⁶.”

3. The sliding process is not clear. How defects, number of layers, diameters, helicities affect the sliding and the polarization is also not clear.

We thank Reviewer 3 for proposing this question. The process of the sliding is an interesting subject to study. The sliding process of interfacial ferroelectricity has been manifested in the 2D van der Waals stacks in previous studies, which involve the immigration of domain wall [Nat. Nanotechnol. 13, 204–208 (2018). Science 372, 1462–1466 (2021). Nat. Nanotechnol. 2022, 17(4): 390-395. Nat. Nanotechnol. 17, 367–371 (2022)]. Unlike the 2D vdW assembly that can expose the surface to the scanning probe microscopy, the 0D interface constructed by the method in this work is secluded and cannot be directly manifested by the scanning probe microscopy. However, the 0D interface could be smaller than the size of the single domain, which might lead to a faster sliding process. In the switch speed test as shown in Fig. R8, therefore it can be deduced that the sliding

process happens in 160 ns or less, which was only limited by the measuring instrument in our lab.

Fig. R8 | The switching speed of the device. a, Programmed and measured waveforms of the electrical pulse. **b,** The high/low resistance states before and after the applied pulse.

We have added the relative discussion to the revised manuscript (line 190 to 196) and we thank the Reviewer 3 for the constructive suggestion.

“Thirdly, the switching speed of the 0D sliding ferroelectric diode is evaluated, which is a key property of performance for memory devices. Programmed electrical pulse is applied to the device with its pulse width experimentally measured, as shown in Supplementary Fig. 13. The programmed pulse width is 140 ns and the measured pulse width is 160 ns, which is the shortest pulse available from our lab. The resistance switch is observed before and after applying the pulse, as shown in Fig. 4d, indicating that the switch happens within the pulse period of ns.”

Further on, we discuss the factors that could potentially affect the sliding and the polarization:

Defects: Defects such as atomic vacancy can transfer by the inducement of electrical field, which was observed at the MoS₂-gold interface and generates the resistance switch, as reported in the previous studies [Nat. Nanotechnol. **10**, 403-406(2015). Nano letters **18**, 434-441 (2018). Nat. Commun. **9**, 2524, (2018). Nat. Nanotechnol. **16**, 58-62, (2021)]. The migrating atomic vacancy that carries net charge can cause a shift of the electrical polarization across the 0D interface. In this work, we have designed and conducted the experiment that excluded the effect from defects, which should not be the major factor that caused the switched polarization in this study. Specifically, we spin-coated PMMA of 300 nm thickness to cover the WS₂ 0D van der Waals junction device, and repeated the electrical measurement. The conductivity of the 0D interface remained at the same level, however, the phenomenon of rectification switch or resistive modulation disappeared. Since the filling PMMA does not block the migration of the defect but prevents WS₂ nanotube from sliding, the polarization shift should result from the mechanical sliding guaranteed by the superlubricity at the vdW interface.

Number of layers and diameter: For the principle of sliding ferroelectricity, only the outermost layers have a direct effect on the polarization of the interface, therefore the inner layers should not be a direct factor that generates the sliding ferroelectricity. However, the number of layers and diameters might have an influence on the strain distribution of the crossbar, which might affect the magnitude of polarization and coercivity as a side effect.

Helicity: The realization of sliding ferroelectricity is dependent on the rhombohedral stacking. Therefore, helicity should be a key factor. The DFT results in Fig. 4d-g and Supplementary Fig. 10

also demonstrate that switchable polarization exists in rhombohedral stacking of WS₂, and is absent in other stacking orders such as parallel and antiparallel stacking [Nat. Nanotechnol. 17, 367-371(2022)].

We thank Reviewer 3 for inspiring the above discussion, we have added related discussion in the manuscript (line 134–line162):

“Here we discuss the mechanism of the polarization switch at the 0D vdW junction. Although the 0D vdW junction shows a ferroelectric diode-like behavior, the source of the polarization shift should be carefully verified. Two possible mechanisms might contribute to such polarization switch: The first hypothesis, as shown in Fig. 3a, is the electric field induced atomic vacancy transfer across the 0D interface, which was observed at the MoS₂-gold interface and generates the resistance switch, as reported in the previous studies⁵¹⁻⁵⁴. The migrating atomic vacancy that carries net charge can cause a shift of the electrical polarization across the 0D interface. The second hypothesis, as shown in Fig. 3b, is that the 0D vdW interface changes its stacking order by atomic scale sliding, which results in interfacial ferroelectricity, reported earlier in subtle 2D vdW interface of hBN, TMDCs, and sandwiched graphene^{17-19,21-29}. Given the infeasibility of the in-situ observation in this case for the potential single atomic vacancy migration or atomic scale sliding, here we have to design an operable experiment for convincing verification. As shown in the upper insert of Fig. 3c, we spin-coated PMMA of 300 nm thickness to cover the WS₂ 0D van der Waals junction device, and repeated the measurement shown in Fig. 2. The conductivity of the 0D interface remained at the same level, however, the phenomenon of rectification switch or resistive modulation disappeared, as shown in Fig. 3c. Since the filling PMMA does not block the migration of the atom vacancy but prevents WS₂ nanotube from sliding, the polarization shift should result from the mechanical sliding guaranteed by the superlubricity at the vdW interface. Therefore, the 0D vdW sliding ferroelectric diode device can be regarded as a special branch of NEMS memory^{55,56}. The general theory has been proposed and well demonstrated^{16,57}. We performed density functional theory (DFT) simulation on the stacked WS₂. The interfacial differential charge densities of rhombohedral stacked WS₂ explicitly show charge redistribution between the top and bottom layers (Fig. 3d, e)²⁵. Their line profiles in Fig. 3f show explicit switched electric polarization at the interfaces, and Fig. 3g shows interlayer potential difference of rhombohedral stacked WS₂, which is absent in parallel and antiparallel stacked WS₂ (Supplementary Fig. 10)²⁶.”

4. The proposed applications are largely speculative.

We thank Reviewer 3 for the comment. This work demonstrate that solid-state ferroelectricity can be scaled down to 0D with the dedicated van der Waals interface in an ultimately small scale. The device of 0D van der Waals interfacial ferroelectric diode shows not only the electrical resistance switch, but also the programmable photovoltaic effect that covers the nearly full visible range. Here we illustrate its significance towards the application in ferroelectric memory and ferroelectric photovoltaics, hoping to address the concern about applications from the Reviewer 3.

Firstly, the down-scaling of ferroelectricity has long been perused by the research community of ferroelectric memory. The size effect of ferroelectricity is a major obstacle to the integration of ferroelectric memory. Previous studies showed that interfacial sliding ferroelectricity exists in two-dimensional stacks. That the solid-state ferroelectricity can be scaled down ultimately to 0D, is not only significant scientific progress that finds the missing part of the ferroelectric paradigm, but also

demonstrates the potential capacity for high-dense integration of ferroelectric memories.

Secondly, previous studies have demonstrated discrete ferroelectric field-effect transistor devices with interfacial sliding ferroelectricity [Nat. Nanotechnol. 17, 367–371 (2022). Nat. Commun. (2022) 13:7696]. The sliding ferroelectricity modulates the electrostatic potential of the channel material and the ON/ OFF state can be distinguished by measuring the conductance of the channel. In this work, the realization of the ferroelectric diode is based on the intuitive resistance switch of the 0D van der Waals interface. This provides the direct operations of read and write, the compact device structure, and the reduced complexity in fabrication processes.

Thirdly, programmable photovoltaics is a key component towards the recent emerging in-sensor computing. For example, a recent work “Ultrafast machine vision with 2D material neural network image sensors” by Lukas Mennel et. al. demonstrated that an image sensor can itself constitute an artificial neural network that can simultaneously sense and process optical images without latency [Nature, 2020, 579(7797): 62-66.]. However, the lack of the memorial function of the photodetector blocks the scaling of the sensor, as stated in the paper:

“Let us now comment on the prospects for scalability. In our present implementation the weights of the ANN are stored in an external memory and supplied to each detector via cabling. **Scaling will require storing the weights locally.** This could be achieved, for example, by using ferroelectric gate dielectrics or by employing floating gate devices.” [Nature, 2020, 579(7797): 62-66]

The programmable photovoltaics realized in this work integrate photodetection and memory in one device based on the 0D interfacial van der Waals sliding ferroelectricity. The programmable photovoltaics achieves the local restore of the photo response as the weights of the ANN, which promotes the scaling of the in-sensor computing.

In all, we agree that the current research does not fulfill the achievement of large-scale ferroelectric electronics, which would be a pursuit of long term for community of ferroelectricity. The demonstration of solid-state ferroelectric down-scaled to 0D, the compact ferroelectric diode thereof, and the programmable photovoltaic is of significance towards the future application together with the previous studies of van der Waals interfacial sliding ferroelectricity. [Nature 560, 336-339, (2018); Nature 588, 71-76, (2020); Science 372, (2021); Science 372, 1458-1462 (2021); Nature, 1-5 (2022); Science 376, 973-978 (2022); Nature 613, 48-52 (2023); Nat. Nanotechnol. 17, 367-371 (2022); Nat. Commun. 13, 7696 (2022); Nat. Commun. 14, 36 (2023)] Here we avoid using arbitrary inference about the application in the conclusion (manuscript, line 244-line 247):

“This work provides the strategy to scale ferroelectricity down to 0D, and demonstrates a branch of device that integrates 0D vdW interfacial ferroelectrics, semiconductor electronics, photovoltaics, and NEMS with potential application to be explored for next-generation versatile electronic systems.”

REVIEWER COMMENTS

Reviewer #1 (Remarks to the Author):

The authors have sufficiently addressed most of my concerns. I can recommend for publication in Nature Communications after the following minor comments are addressed.

1. Switching Speed: Figure 4d & S13

To prove ferroelectric switching at high speeds, the resistance data before/after the applied pulse needs to be shown in a smaller time resolution. Currently, the x-axis in Figure 4d is in seconds. It should be in nanoseconds, i.e. the same x-axis resolution of Fig. S13 for the voltage waveform.

2. Endurance

Also, repeated ferroelectric resistive switching should be shown in a pulsed manner. For example, the high resistance and low resistance states (y-axis) should be plotted against number of cycles (x-axis) for a fixed pulse amplitude, pulse width, and read voltage.

The current endurance data to 10 cycles (Fig 2c, Fig S9) is insufficient, and it is from DC voltage sweeps, not pulsed measurements. Please cycle the device to its breakdown in a pulsed manner and report the number of cycles to breakdown.

3. Word choice

Main Text Line 35-37: The ultrathin doped hafnium/zirconium oxide references cited are not amorphous, so please remove the word "amorphous" from the text. And it is unclear what the word "subtle" refers to in "subtle van der Waals stacked assembly. Perhaps replace "subtle" with the word to "twisted" or "sliding" or something more appropriate. Or simply remove the word "subtle."

Reviewer #2 (Remarks to the Author):

The authors addressed all the concerns, providing more data than required. It is ready for publication

Reviewer #3 (Remarks to the Author):

The authors have satisfactorily addressed all the concerns raised in my review report. I have no more comments and suggestions.

Reviewer#1 (Remarks to the Author):

The authors have sufficiently addressed most of my concerns. I can recommend for publication in Nature Communications after the following minor comments are addressed.

We thank Reviewer 1 for the effort in reviewing the revised version of the manuscript, which helps us to further improve the quality of the manuscript. Our responses to the 2nd round of comments are below.

1. Switching Speed: Figure 4d & S13

To prove ferroelectric switching at high speeds, the resistance data before/after the applied pulse needs to be shown in a smaller time resolution. Currently, the x-axis in Figure 4d is in seconds. It should be in nanoseconds i.e. the same x-axis resolution of Fig. S13 for the voltage waveform.

We thank Reviewer 1 for proposing this change. We understand that the Reviewer 1's concern is about the switching speed, or how fast the transition can be completed. In the previous test, we used the electrical pulse with a width of 160 ns to switch the resistive status. It was confusing that the results in Figure 4d are presented in seconds, rather than nanoseconds.

We would like to explain here that the results presented in Figure 4d are the electrical measurements of the four probes before and after applying the short electrical pulse. The change of the resistive states indicates that the transition happens within the duration of the pulse, which should be within hundreds of nanoseconds or shorter. In other words, the pulse with the width of 160 ns is sufficient to make the switch. Such a “before and after” measurement strategy was adopted when the sampling time required is larger than the pulse width [for example, see Tsai et al., Device and Materials Reliability, IEEE Transactions, 5, 217-223, 2005; Min-Kyu Kim et al., Sci. Adv. 7, eabe1341, 2021; Wu et al., Adv. Mater. 31, 1806790, 2019; Nature Communications, 14, 2757, 2023]. We have revised Figure 4d accordingly for the illustration of the process with an insert of the pulse.

Fig. R1 | The high/low resistance states before and after the applied pulse. Insert: The programmed and measured pulse waveform.

As suggested by the Reviewer, a temporally resolved switching speed would present the switching process more intuitively. However, we were unable to obtain the temporally resolved switching process within the pulse, due to the limitation of the sampling speed of the equipment available in our lab. The four-terminal measurement adopted in this work includes the current sampling of the OD junction at the nA level as well as the voltage sampling of the probes, which require the suppression of the current noise level much lower than the current that goes through the junction. The current noise level of different sampling speeds of the equipment is shown in Fig. R2. Low-precision sampling can improve the measurement speed, but the magnitude of the current noise significantly increases and would be close to the magnitude of the current passing through the junction. Therefore, we had to adopt the lower sampling speed for the improved precision.

Fig. R2 | Current noise level with different sampling time.

2. Endurance

Also, repeated ferroelectric resistive switching should be shown in a pulsed manner. For example, the high resistance and low resistance states (y-axis) should be plotted against number of cycles (x-axis) for a fixed pulse amplitude, pulse width, and read voltage.

The current endurance data to 10 cycles (Fig 2c, Fig S9) is insufficient, and it is from DC voltage sweeps, not pulsed measurements. Please cycle the device to its break down in a pulsed manner and report the number of cycles to breakdown.

We thank Reviewer 1 for raising this issue. In the previous revision, we showed that the switched states of high and low resistance have long retention and stability over time. We understand that the new concern is about the endurance of cycles between the high and low resistance states (fatigue).

We cycled the device as suggested and compare the results with previous studies, as shown in Fig. R3 and Table R1, respectively. The device shows good endurance in the initial few cycles (10 V, 350 ns). However, we found the device stuck at the high resistance status, or fatigue of the device. We tried to increase the amplitude of the electrical pulse, but the increased amplitude of 15 V resulted in breakdown of the device, as shown in Figure R4.

Fig. R3 | Endurance of resistive switching with the pulses. The amplitude of the pulse is ± 10 V. Read voltage ~ 1 V.

Materials	Dimension	DC	Times demonstrated	Pulse	Times demonstrated	Reference	Note
h-BN	2D	Y	>4	Y	11	Kenji Yasuda. et al. Science 372,1458-1462(2021)	-
h-BN	2D	Y	1	N	-	M. Vizner Stern et al., Science 372,1462-1466(2021)	-
T_d -MoTe ₂	2D	Y	10	Y	3	Apoorv Jindal. et al. Nature 613, 48-52 (2023)	-
MoS ₂ /WS ₂	2D	Y	about 36	Y	40-60	Lukas Rogee. et al. Science 376,973-978(2022)	current leakage overtook
WSe ₂	2D	Y	1 000	N	-	Yang Liu. et al. Nano Lett. 2022, 22, 3, 1265-1269	-
MoS ₂	2D	Y	>5	Y	9	Astrid Weston. et al. Nat. Nanotechnol. 17,390-395 (2022)	-
WSe ₂ and MoS ₂	2D	Y	1	N	-	Swarup Deb. et al. Nature 612, 465-469 (2022)	-
WTe ₂	2D	Y	9	N	-	Zaiyao Fei. et al. Nature 560, 336-339 (2018)	-
WSe ₂ , MoSe ₂ , WS ₂ and MoS ₂	2D	Y	>4	N	-	Xirui Wang. et al. Nat. Nanotechnol. 17, 367-371 (2022)	-
γ -InSe	2D	Y	4	N	-	Fengrui Sui. et al. Nat Commun 14, 36 (2023)	-
3R-MoS ₂	2D	Y	40	N	-	Peng Meng. et al. Nat Commun 13, 7696 (2022)	-
graphene	2D	Y	20	N	-	Zhiren Zheng. et al. Nature 588, 71-76 (2020)	-
WS ₂	0D	Y	10	Y	About 20	This work	-
GaSe	2D	Y	tens of times	Y	5000	Wenhui Li. et al. Nat Commun 14, 2757 (2023)	intralayer sliding

Table R1 | Endurance of ferroelectric devices.

The endurance of tens of cycles at the current stage is not a satisfying endpoint. Previous studies have reported the endurance of different sliding ferroelectricity devices from a few cycles and up to thousands of cycles, as shown in Table R1. The fatigue failure of traditional ferroelectric materials has been long explored, which could be the domain wall pinning, domain nucleation suppression, microcracks, and accumulation of the space or injected charge. [Nature communications, 12(1), 2095, 2021; Reports on progress in physics, 61(9): 1267, 1998; Journal of Applied Physics, 105(2), 024101, 2009; Journal of Materials Research, 30(3): 364-372, 2015] It is not clear yet what is the mechanism of the fatigue of sliding ferroelectricity in the present work. This point could be explored with the recently developed operando electron microscopy

investigation [Nature Materials, 2023, in print].

Fig. R4 | Optical microscope and scanning electron microscope images. a, b, Optical microscope images of the device before and after the breakdown. **c,** Scanning electron microscope image of the part broken down.

We have supplemented the above discussion in the revised version of the manuscript (line 199-206) and supplementary information (Supplementary Fig. 15, line 89-91).

“However, the endurance of the device versus pulses is less satisfying. Supplementary Fig. 15 shows the cycles of switches of the 0D sliding ferroelectric diode. The device presents tens of switching cycles before the fatigue. The fatigue failure of traditional ferroelectric materials has been long explored, which could be the domain wall pinning, domain nucleation suppression, microcracks, and accumulation of the space or injected charge⁵⁹⁻⁶². It is not clear yet about the mechanism of the failure of sliding ferroelectricity. It could be possibly explored with the help of the recently developed operando electron microscopy investigation⁶³.”

3. Word choice

Main Text Line 35-37: The ultrathin doped hafnium/zirconium oxide references cited are not amorphous, so please remove the word "amorphous" from the text. And it is unclear what the word "subtle" refers to in "subtle van der Waals stacked assembly". Perhaps replace "subtle" with the word to "twisted" or "sliding" or something more appropriate. Or simply remove the word "subtle".

We thank Reviewer 1 very much for the suggestion, we have modified the text in the revised manuscript:

“For example, the recent research on two-dimensional (2D) ferroelectricity has brought the thickness down to the atomic limit, including the ultrathin doped hafnium/zirconium oxide⁸⁻¹⁰,”

and

“the monolayer ferroelectric materials¹¹⁻¹⁵, and the van der Waals (vdW) stacked assembly¹⁶⁻
30.”

REVIEWERS' COMMENTS

Reviewer #1 (Remarks to the Author):

The authors have satisfactorily addressed all the concerns raised in my review report. I can recommend for publication in Nature Communications.

One final small comment: I suggest the authors include Table R1 in the supplementary materials and reference it in the recently added endurance discussion in the main text (lines 199-206). Significant work needs to be done to address endurance (fatigue) behavior in sliding ferroelectricity, so Table R1 is helpful to emphasize this point. This change does not require further review.

Reviewer #1(Remarks to the Author):

The authors have satisfactorily addressed all the concerns raised in my review report. I can recommend for publication in Nature Communications.

One final small comment: I suggest the authors include Table R1 in the supplementary materials and reference it in the recently added endurance discussion in the main text(lines199-206). Significant work needs to be done to address endurance(fatigue) behavior in sliding ferroelectricity, so TableR1 is helpful to emphasize this point. This change does not require further review.

We thank Reviewer 1 very much for the suggestion, we have added the Table R1 in the revised Supplementary Information and modified the text in the revised manuscript.